

# Managed black truffle-producing systems have greater soil fungal network complexity and distinct functional roles compared to wild systems

Vasiliki Barou[1], Jorge Prieto-Rubio[2], Mario Zabal-Aguirre[3], Javier Parladé[1], Ana Rincón[4]

[1]IRTA, Sustainable Plant Protection, Centre de Cabrils, 08348, Cabrils, Barcelona, Spain

[2]Centro de Investigaciones Sobre Desertificación (CIDE), CSIC-Universidad de Valencia-Generalitat Valenciana, 46113, Moncada, Valencia, Spain

3Departamento de Biología, Universidad Autónoma de Madrid, 28049, Madrid, Spain

[4]Instituto de Ciencias Agrarias, ICA-CSIC. C/ Serrano 115bis, 28006, Madrid, Spain

*Correspondence to*: Ana Rincón (ana.rincon@csic.es)

**Abstract.** Black truffle (*Tuber melanosporum* Vittad.), a valued edible fungus, has been thoroughly studied for its ability to modify soil conditions and influence microbial communities in its environment as it dominates the space. While direct associations of black truffle with microbial guilds offer insights into its competitiveness, the role of these interactions in ecosystem functions remain unclear. This study aims to assess the patterns of soil fungal community within the black truffle brûlés across different producing systems (managed *vs* wild) and seasons (autumn *vs* spring), to determine the role *of T. melanosporum* in the structure of the fungal networks, and to identify the contribution of main fungal guilds to soil functioning in these systems. To address this, network analysis was employed to construct the fungal co-occurrence networks in the brûlés of black truffle plantations and wild production areas in forests. Black truffle plantations showed greater fungal homogeneity, network complexity and links compared to forests, indicating enhanced stability, possibly due to reduced plant diversity and uniform conditions, while seasonality did not affect the fungal network structure. Despite its dominance in the brûlés, *T. melanosporum* was not a hub species in neither truffle-producing systems and exhibited few interactions, mainly with saprotrophs and plant pathogens. Saprotrophic fungi, with partial contributions from ectomycorrhizal and plant pathogen guilds, were the key contributors to carbon and nutrient cycling in both systems. These results improve our understanding of the ecology, biodiversity and functioning of black truffle-dominated soils that could enable more effective management strategies in black truffle plantations.

## 1 Introduction

Black truffle (*Tuber melanosporum* Vittad.) is a valued edible fungus naturally produced in Mediterranean ecosystems, where it forms ectomycorrhizal associations with host plants like *Quercus* spp. Due to its high market value, this fungus has been cultivated beyond its natural distribution area (Reyna and Garcia-Barreda, 2014), and its production has promoted the use of abandoned agricultural lands and contributed in maintaining the soil quality (Leonardi et al., 2021). Either grown in



forests or plantations, the black truffle mycelium dominates the area around the colonised trees, reducing the vegetation and giving the appearance of a burnt area called "brûlé" (Streiblová et al., 2012; Schneider-Maunoury et al., 2018).

The colonisation and production success of black truffle has been extendedly studied in relation to the abiotic environment
and its effects on the fungus´ growth (García-Montero et al., 2007; Garcia-Barreda et al., 2019; Barou et al., 2024). In the light of the niche construction theory, where black truffle is thought to have the ability to modify the surrounding abiotic edaphic conditions (Bragato, 2014; García-Montero et al., 2024), soil biotic components can also be affected by black truffle in a shared niche (Mello et al., 2015). Under the same soil environment, in a process of dominating the space, the black truffle may show antagonistic and/or synergistic interactions with the surrounding microbial communities. For instance, a
competitive interaction has been found between *T. melanosporum* and other ectomycorrhizal fungi (ECM) for host plant and niche colonisation in brûlés (Belfiori et al., 2012; De Miguel et al., 2014), as well as with other truffle species (Marjanović et al., 2020). Further potential competitive effects have been also observed with arbuscular mycorrhizal fungi (AMF) (Barou et al., 2023), molds, yeasts and plant pathogens, whose abundances have been negatively associated with that of *T. melanosporum* over time (Oliach et al., 2022).

Although the direct associations of black truffle with other microbial guilds have provided some valuable insights regarding its competitiveness, there is still a gap of knowledge on what is their role in the ecosystem functions. Previous work indicated that soil microbial shifts might have functional consequences, affecting important ecosystem processes (Pérez-Izquierdo et al., 2017). For example, the competition between ECM and saprotrophic fungi in forest soils for water and nutrients has been considered to suppress decomposition rates, leading to greater carbon sequestration (Bödeker et al., 2016;
Fernandez and Kennedy, 2016). However, this phenomenon is not consistent across the world's biomes, as it depends on several environmental and anthropogenic factors (Fernandez and Kennedy, 2016; Choreño-Parra and Treseder, 2024). In this sense, the projected increased temperatures and decreased precipitations due to global warming (Guiot and Cramer, 2016) are expected to negatively impact the performance of vegetation and associated microorganisms, especially in highly water-limited ecosystems, such as the Mediterranean ones (Querejeta et al., 2021). In this context, since ECM fungi are the main
receivers of photosynthetic carbohydrates from the host (Itoo and Reshi, 2013), their reduction could accelerate the carbon cycling stimulating saprotrophic activity (Fernández and Kennedy, 2016). It is, therefore, important to study the associations between the soil fungal diversity and functionality considering the particular climate and soil conditions of a given ecosystem.

As it concerns the black truffle producing systems, seasonality can also influence the competitiveness of black truffle and
soil microbial shifts, since the dynamics of the fungus change throughout its long life-cycle (Garcia-Barreda et al., 2020), and soil microbiota fluctuates across the seasons as a consequence of temperature, soil moisture and nutrient variation (Koranda et al., 2013; Luo et al., 2019), ultimately impacting soil functionality (Vořiškova et., 2014; Siles and Margesin, 2017; Júnior et al., 2019; Wang et al., 2024a).

Considering all the above, the research on diversity and dynamics of soil microbial communities, together with the
underlying ecological factors, is essential to better understand their functions (Orgiazzi et al., 2012) and to finally



ameliorating black truffle production (Antony-Babu et al., 2013). In this sense, the use of network analysis in soil ecology research has increased substantially over the past decade (Guseva et al., 2022). This approach considers taxa that co-occur simultaneously at the same place as being connected by links, ultimately forming networks (Goberna and Verdú, 2022). Analysing these networks has been considered to provide valuable insights into microbial distribution across multiple sites

(Wagg et al., 2019; Goberna and Verdú, 2022; Prieto-Rubio et al., 2024), and to allow the exploration of ecological community complexity (Guseva et al., 2022). Recently, Barou et al. (2025) have shown that, together with specific soil properties, black truffle abundance and Ascomycetes richness predict soil enzymatic activity in black truffle plantations. Indeed, the biodiversity and composition of biological communities are inherently associated with soil functioning (Wagg et al., 2021; Prieto-Rubio et al., 2024). In black truffle-dominated systems, the species co-occurrence network analysis may

contribute to deciphering the patterns of fungal community composition, understanding their complexity, thus allowing a deeper understanding of soil functioning (Barberán et al., 2012; Guseva et al., 2022).

Given the potential habitat effect associated with different land uses on soil microbial community structure (Byers et al., 2024; Piñuela et al., 2024) and functioning (Flores-Rentería et al., 2018; Barou et al., 2025), the study of co-occurring taxa in managed *vs* wild truffle-producing environments may unravel the intricate ecological dynamics of truffle-dominated brûlés.

More specifically, the variation of environmental conditions among these black truffle-producing systems may be linked to variations in functional fungal guilds that could play an important role in organic matter decomposition and nutrient´s mobilisation (Banerjee et al., 2016). This role, however, is often studied individually for specific guilds while ignoring others inhabiting in the same environment (Fernandez and Kennedy, 2016). Studying soil carbon and nutrient cycling in relation to the whole fungal biodiversity is crucial for understanding the nutritional feedbacks of the plant-soil system and for

optimising management strategies of truffle-producing plantations.

In a previous work, we have shown that black truffle abundance negatively impacts soil enzymatic activity (Barou et al., 2025). We now aim to go a step further and uncover the organisation of soil fungal communities, in both managed and wild black truffle-producing systems, to understand the role of *T. melanosporum* in structuring them, and how this translates into soil functioning. In terms of practical implementation, the precise microbial structure and the contribution of fungal

communities to soil functioning in these systems can valuably inform the development of optimal biofertilisation regimes for truffle cultivation. The specific objectives of this study were 1) to analyse the effect of black truffle-producing ecosystem (managed *vs* wild), and seasonal effects (spring *vs* autumn) on soil fungal communities, and to decipher the role of *T. melanosporum* structuring them; and 2) to identify the main functional fungal guilds operating in soils of managed and wild truffle dominated systems, and their relative contribution to soil carbon and nutrient cycling.

Considering that microorganisms and plants have interacted over long evolutionary time in wild ecosystems (Song et al., 2019; Pérez-Lamarque et al., 2022), and the less predictable environmental conditions of wild than managed ecosystems, we expected soil fungal communities to be richer and more complex in forests than in plantations. Moreover, a differential seasonal impact on soil fungal communities of both truffle-producing systems was also expected (hypothesis 1). Given the strong black truffle effect inside the brûlé (Streiblová et al., 2012; Barou et al., 2025), we additionally envisaged that *T.*



*melanosporum* would markedly contribute to the soil fungal network structure acting as a hub species within the community, i.e., highly connected with other fungi (hypothesis 2). Hub species connecting multiple species in an ecosystem may play a key role in energy flow, meaningfully affecting biodiversity and productivity (Toju et al., 2018). Given the distinct organic inputs in wild and managed truffle systems (Barou et al., 2024; Barou et al., 2025), and the distinct dominant mycorrhizal plant species in forests and agricultural systems postulating key microbial functional shifts and nutrient turnover (Phillips et

al., 2013), we anticipated finding different representative ecological fungal guilds (i.e., fungi with distinct lifestyle) within the fungal networks in each system. Specifically, we expected a greater prevalence of ECM *vs* saprotrophs in the fungal network of forests compared to that of plantations, each differentially impacting the soil functioning (i.e., carbon and nutrient cycling) in the respective black truffle-producing systems (hypothesis 3).

To test these hypotheses, we first investigated the diversity and species composition of fungal communities in soils collected

inside the brûlé of black truffle productive oaks in both, forests and managed plantations, and analysed the relative abundance of representative fungal guilds (i.e., saprotrophs, mycorrhizal, pathogens, parasites). We further built fungal co-occurrence networks and studied their properties and the role of *T. melanosporum* within their complexity, as a function of the study factors. Finally, we analysed the net contribution of different fungal guilds within the networks to soil carbon and nutrient cycling (i.e., data of eight potential soil enzymatic activities), within the brûlé of black truffle producing forests and

plantations.

## 2 Materials and Methods

### 2.1 Experimental design, soil sampling and processing

Eighteen sites along the natural distribution of the black truffle in Spain were selected. Per each site, two types of truffle-producing systems were considered: plantations (17 sites) and forests (10 sites) (Fig. S1). All plantations are effective truffle

producing holm oak monocultures, more than 10-years-old, where standard agriculture management practices for truffle culture (mainly irrigation, tilling and pruning) are regularly conducted. In forests, *Quercus ilex* L. dominates the uppermost canopy layer. The study sites have continental to Mediterranean climate, with seasonal variations in water availability and a mean altitude of $936 \pm 251$ (SD) m (a.s.l.) (for more details see Barou et al., 2024 and Barou et al., 2025).

Four truffle-productive (according to local owners) *Q. ilex* trees were selected per site, and a total of 116 trees (68 in

plantations and 48 in forests) were sampled in autumn 2019 and spring 2020. A metallic probe (4 cm diameter x 20 cm) was used to extract soil samples from the four orientations between the tree trunk and the limit of the brûlé, which were pooled into a unique composite soil sample per tree. A total of 232 soil samples were collected and stored at 4 °C until processing.

Soil samples were measured in previous works for physical-chemical properties (Table S1; modified from Barou et al., 2025). Soil functioning was proxy through the potential activities of eight exoenzymes related with the carbon (β-

glucosidase, β-cellobiohydrolase, β-xylosidase, β-glucuronidase and laccase), nitrogen (chitinase and leucine-aminopeptidase) and phosphorous (alkaline phosphatase) cycling, as previously detailed in Barou et al. (2025). Briefly, soil





samples (1 g) were incubated overnight in specific buffers at 25°C. Enzymatic activity was then determined using fluorogenic substrates or by a photometric assay (laccase), with a plate reader at specific excitation/emission wavelengths, and the results were normalized to soil and organic matter content.

## 2.2 Molecular and bioinformatics analysis

Soil DNA was extracted and metabarcoding sequencing (Illumina MiSeq) performed by targeting the fungal ITS1 region, as described in Barou et al. (2025), resulting in a final dataset of 231 samples, after a sequencing failure in one sample. Briefly, the DNA was extracted from 0.25 g of sieved soil using the DNeasy PowerSoil Pro kit, the fungal diversity was assessed through metabarcoding using the primers ITS1F/ITS2 (Gardes and Bruns, 1993), and blank controls were included in both assays to monitor possible contaminations.

Bioinformatics was conducted with the DADA2 pipeline (Callahan et al., 2016; R Core Team, 2022). Briefly, after primer's removal, sequences were filtered and merged, taxonomically assigned, and clustered into OTUs (97% similarity), according to the v8.2 UNITE database. A final matrix of 8,654 fungal OTUs and 7,151,228 reads was obtained, after removing OTUs with less than five reads. Once the taxonomic assignations were obtained, OTUs closely related to a recognised fungal guild (i.e., saprotrophs, ECM, AMF, plant pathogens, parasites, endophytes, epiphytes or lichens) were identified by the FUNGuild and FungalTraits databases (Tedersoo and Smith, 2013; Tedersoo et al., 2014; Nguyen et al., 2015; Põlme et al., 2020).

Before any further analyses, rarefaction was applied to check for suitable sequencing depth across all samples (Fig. S2), and the whole database was normalised by dividing the number of sequences (i.e., reads) per OTU in a sample by the total number of sequences of that sample.

## 2.3 Network analysis

To further investigate the structure of soil fungal communities, co-occurrence network analysis was performed following the methodology described by Prieto-Rubio et al. (2024). For this, the OTU's abundance matrix was first rescaled by the respective minimum sequencing depth for each dataset (i.e., whole dataset, and datasets separated by plantation or forest samples). Low frequent OTUs that appeared in <10 % of the samples in the respective datasets were discarded (Wagg et al., 2019). Then, to generate the fungal co-occurrence networks, the SPIEC-EASI algorithm was applied to each dataset with the spiec.easi function in the "SpiecEasi" R package (Kurtz et al., 2015). This method seeks to detect a link (i.e., co-occurrence) between any two nodes (i.e., two OTUs), assuming that their abundances are not independent and thus, discarding spurious associations between OTUs (Kurtz et al., 2015). The construction of networks was based on the Meinshausen and Bühlmann (2006) neighbourhood selection method. The lambda ratio was set to 0.01 and, to detect the most parsimonious network, 50 values of lambda for every 100 cross-validation permutations were fitted by using the Stability Approach to Regularisation Selection (StARS) (Liu et al., 2010). The inference of the fungal networks and their preparation for visualisation were performed as described in Birt and Dennis (2021). The resulting fungal networks were graphically represented using the



Gephi software v0.10 (Bastian et al., 2009), applying both the Fruchterman-Reingold distribution and ForceAtlas2 attraction-repulsion algorithms in the simulations (Fruchterman and Reingold, 1991; Jacomy et al., 2014).

To measure the contribution of each OTU – including that of *T. melanosporum* – in the network structure, network centrality metrics (degree, eigenvector, closeness, and betweenness centralities; Brede, 2012) and the hub score metric as proxy of keystone taxa (Deguchi et al., 2014), were calculated in Gephi. In addition, the modularity algorithm proposed by Blondel et al. (2008) was also run in Gephi to determine the fungal network divisions into sub-communities, i.e., modules characterised by dense links inside the community compared to the sparser links with other communities (Newman, 2006). The nodes (equivalent to OTUs) belonging to the same module were assigned a distinctive colour, and the node size was determined by the hub score of that OTU. On the other hand, the links between the nodes were coloured according to the type of association, i.e., positive or negative. To unravel the black truffle interactions with the fungal community, the networks were finally filtered by the links established only with *T. melanosporum* and positive and negative co-occurrences with other fungi were identified.

## 2.4 Statistical analysis

In order to assess the effects of the type of truffle production system (hereafter "type") and different seasons of sampling (hereafter "season") on soil fungal diversity (hypothesis 1), a linear mixed model (LMM) was first run, with OTU richness (α-diversity) as response variable, type and season as fixed factors (including their interaction), and the site as random factor. The LMM was performed with lmer function from "lme4" R package (Bates et al., 2015) and pairwise comparisons to find significant differences between the different groups were conducted with lsmeans (Lenth, 2016). β-diversity of soil fungal communities was then calculated based on the Bray-Curtis OTU's dissimilarity matrix, and nonmetric multidimensional scaling (NMDS) was used to draw the compositional differences within the respective type and season treatments, with the metaMDS function from "vegan" R package (Oksanen et al., 2022). Statistical significance of factors was determined by permutational multivariate analysis of variance (PERMANOVA, permutations = 9,999) with the Adonis function using the same Bray-Curtis OTU's dissimilarity matrix. In addition, the vectors of pH, electric conductivity (EC), organic matter (OM), active carbonate, and N, P, K, Mg and Fe content were also fitted in the NMDS with the envfit function to check for correlations.

To investigate the impact of the factors type and season on the co-occurrence soil fungal network (hypothesis 2), we first calculated different structural network properties i.e., OTU richness, the number of links between OTUs and the network complexity (ratio of links to OTU richness) using the total dataset ($N = 231$ samples). The structural network properties were used as response variables in LMMs built with the same fix (type, season) and random (site) factors as described above.

Then, co-occurrence networks were separately built for plantations ($N = 135$) and forests ($N = 96$) datasets. To evaluate the contribution of *T. melanosporum* in the fungal network structure and identify potential hub taxa, besides each OTU hub-score, a comparative ranking list among OTUs was calculated (adapted from Brandon-Mong et al., 2020). Briefly, each OTU was first assigned to a rank based on the values of each of its network centrality metrics (degree, eigenvector, closeness,



betweenness); then, the rankings of all metrics were summed for each OTU, resulting in a final total rank assigned to each OTU. The top-10 highly ranked OTUs were considered as hub species, and the position of *T. melanosporum* within this list was used to evaluate its contribution to the soil fungal network structure. In addition, LMMs were built to explore its

contribution in the network complexity, using the abundance of *T. melanosporum* as a fixed factor and the site as random for plantations and forests, separately.

To test the hypothesis 3, the effect of factors type and season on the most representative ecological fungal guilds (i.e., saprotrophs, ECM, AMF, plant pathogens, parasites) and *T. melanosporum* was first analysed by LMMs, with the relative abundance of each guild as response variable and the model syntaxes similar to that previously explained. Then, to evaluate

whether fungal guilds explained soil carbon and nutrient cycling, we performed a model training by the elastic net regularization method (hereafter as ENET model, Zou and Hastie, 2005). This method incorporates least absolute shrinkage and selection operator (LASSO) and Ridge penalty-based regression modelling, that together allowed to determine how each OTU contributed to predict soil enzymatic activities, by avoiding overfitting and correlation effects between OTUs, and the influence of those that weakly explained variations in enzymatic activities (Zou and Hastie, 2005). For model training, we

standardized fungal OTU matrices (plantation/forest) through the decostand function in the "vegan" R package. These matrices were fitted as predictors of soil enzymatic activities (response variables) by 100-fold simulated regressions across 100 alpha-penalty parameters by the optEnet function in the "glmnet" R package. The significant effects of OTUs on each soil enzymatic activity were determined by standardized effect size (SES) from the selected regression model. SES > 1.96 indicated OTUs that positively explained enzymatic activities, while SES < 1.96 did it in OTUs with negative relationships

with enzymatic activities. These values allowed us to record all OTUs that predicted soil enzymatic activity and classed by forest/plantation and by functional group (Wagg et al., 2019; Prieto-Rubio et al., 2024).

The graphical representation of results (i.e., α-diversity, abundances of ecological fungal guilds and most abundant fungal taxa) was performed with the "microeco" R package (Liu et al., 2021). All analyses were performed with the R software version 4.2.1 (R Core Team, 2022).

## 3 Results

### 3.1 Diversity of soil fungal communities

A total of 8,654 fungal OTUs were identified, with an average of 325 ± 77 (SD) OTUs per sample. The 38.2 % of all OTUs ($N$ = 3,309) – that were also the most abundant – were shared in plantations and forests, and close to 30 % of OTUs (low abundant) were uniquely found in each respective truffle producing system (Fig. 1a). Approximately half of the fungal OTUs

(the most abundant) were found in both seasons, while 21-25 % of them were only found either in autumn or spring, but their relative abundances were just 1-2 % of the total (Fig. 1a). Across the soil samples from the two truffle-producing systems and seasons of sampling, the most abundant phylum was Ascomycota, representing 64-73 % of the total fungal abundance, followed by Basidiomycota and Mortierellomycota, with 15-26 % and 5-8 % relative abundance, respectively (Fig. 1b).





*Tuber melanosporum* accounted for 13-22 % of the total Ascomycota abundance, depending on the treatment (Fig. 1b).

Among the assigned fungal genera in the total OTU dataset (71 % of OTUs), the top-ten most representative included *Tuber*, as the most abundant one in both types of truffle-producing systems and both seasons, followed by *Mortierella*, *Solicoccozyma* and *Fusarium* in plantations and *Mortierella*, *Tomentella*, *Inocybe* and *Picoa* in forests (Fig. S3).

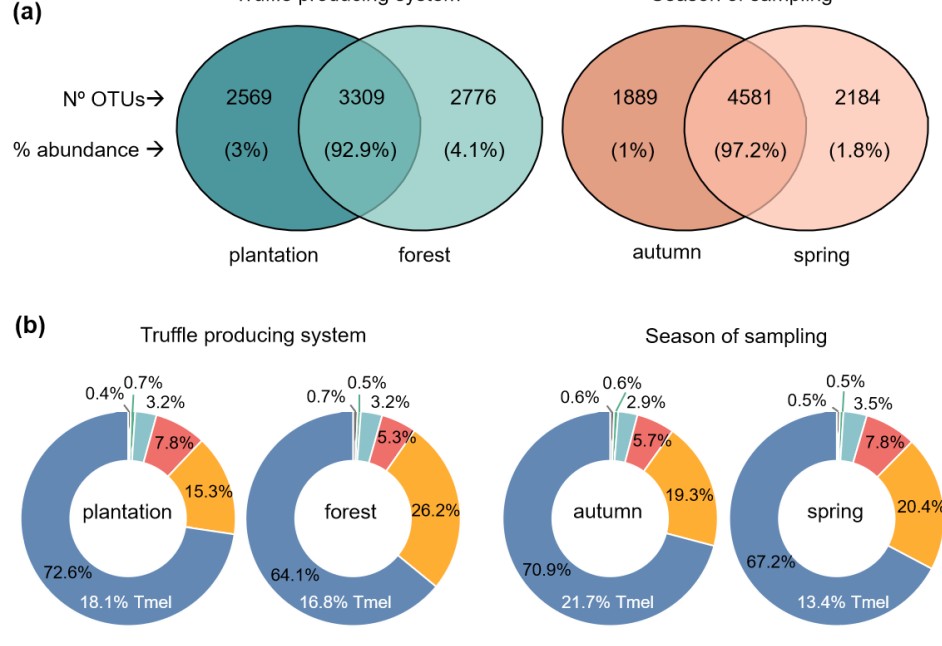

**Figure 1: (a) Sequencing yields and (b) main taxonomic groups of soil fungal communities, depending on the factors studied: type**
**of black truffle producing system (plantation, forest) and season of sampling (autumn, spring). In (b) percentages represent the mean relative abundance of each phylum and the proportion of *Tuber melanosporum* (Tmel) abundance. NA are not taxonomically identified fungal OTUs. *N* = 231 samples, total fungal OTUs = 8,654.**

Contrary to that expected (hypothesis 1), no significant differences in fungal α-diversity were observed between truffle-producing plantations and forests ($p = 0.142$), and only marginal differences due to season were observed ($p = 0.072$) (Fig.

S4a). No interaction among factors was detected. However, when β-diversity was analysed, significantly dissimilar fungal communities were found within the treatments of the respective factors studied: type (F = 13.05, $p < 0.001$, $R^2 = 0.053$) and season (F = 3.8, $p < 0.001$, $R^2 = 0.016$), and more homogeneous fungal communities were found in plantations than in forests, and in autumn than in spring (Fig. S4b). Fungal β-diversity of plantations was correlated with pH, K and active carbonates, while in forests it was mainly correlated with OM, EC, N, Fe and Mg (Fig. S4b). In terms of seasonal patterns,

the fungal community likely correlated with EC in autumn and N, and with active carbonates in spring (Fig. S4b).

To build the overall co-occurrence fungal network, only frequent fungal OTUs present in at least 10 % of samples were retained, and the total fungal network (*N* = 231 samples) was composed of 706 co-occurring OTUs that showed 4,512 links



among them. The type of truffle-producing system did significantly affect the structural properties of the network and, contrary to that expected (hypothesis 1), more complex fungal network and more links between taxa were observed in

plantations than in forests (Fig. 2, Table S2). No effect of season was observed on the fungal network properties (Fig. 2, Table S2).

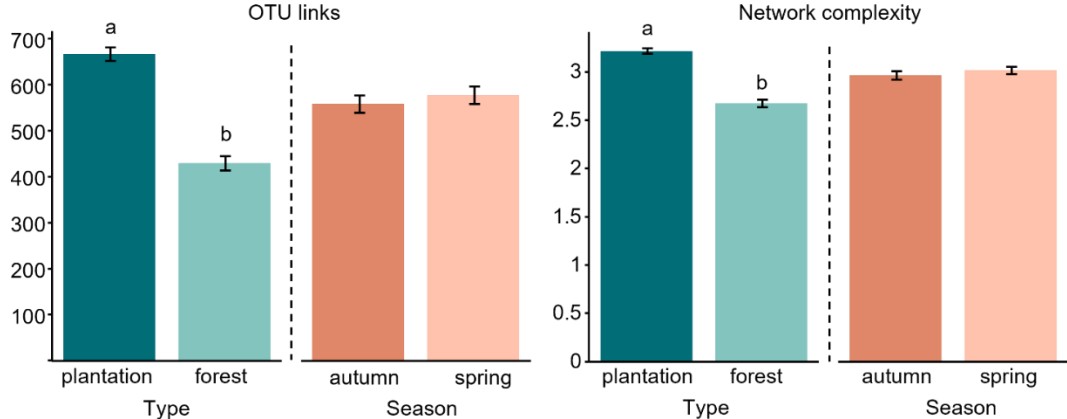

**Figure 2: Structural properties of soil fungal networks in two types of black truffle producing systems (plantation, forest) and two seasons (autumn, spring): (a) OTU links = number of co-occurrences between OTUs, and (b) network complexity = ratio between**
**the number of links and OTU richness. Bars represent the mean ± SE of each variable (*N* samples = 231; OTUs = 706, present in >10% of samples) and letters indicate significant differences between the treatments of a given factor, derived from linear mixed models and pairwise comparisons (see Table S2).**

Regarding the node centrality metrics, the top-10 taxa with the highest total ranking in all network metrics (degree, eigenvalue, closeness and betweenness) included nine Ascomycota (*Stachybotrys chartarum* Corda., *Setophaeosphaeria*

*badalingensis* Crous & Y. Zhang ter, *Arthrinium* sp., *Chaetosphaeronema* sp., *Penicillium jensenii* K. V. Zaleski, *Humicola fuscoatra* Traaen, *Botryotrichum atrogriseum* J. F. H. Beyma, *Microdochium novae-zelandiae* Hern.-Restr., Thangavel & Crous, and one unclassified Ascomycota OTU), and one Mortierellomycota OTU (*Mortierella* sp.). Half of these fungi are classed as litter/soil saprotrophs, while the primary lifestyle of the other half is plant pathogens i.e., *S. badalingensis*, *Arthrinium* sp., *M. novae-zelandiae*, *Chaetosphaeronema* sp. (Põlme et al., 2020). As it concerns the contribution of *T.*

*melanosporum* in the overall soil fungal network structure, when considering the sum of network properties (degree, eigenvector, closeness, and betweenness), it was in the 163[rd] position (out of 706), i.e., within the first 23 % of OTUs in network metrics' ranking, and dropped to 37 % (261[st] position) when only the OTU's hub score (proxy of keystone taxa) was considered. These results seemed to indicate that, contrary to that expected (hypothesis 2), *T. melanosporum* was not likely acting as a hub species structuring the total co-occurring fungal network inside the brûlé.

**3.2 *Tuber melanosporum* representativeness**

To investigate more closely the representativeness of *T. melanosporum* in each type of truffle-producing system, separate network analyses were further conducted for plantations and forests. The resulting fungal networks were composed of 728



and 641 co-occurring OTUs in plantations and forests respectively, which were distributed across 13 different modules in both cases (Fig. 3). Each module was composed of at least 25 OTUs and some modules were more central than others (i.e., modules with higher centrality metrics are more relevant and connected within the network), both in plantations (e.g., modules 3 and 6; PERMANOVA, $p = 0.003$, $R^2 = 0.02$) and in forests (e.g., modules 0, 3 and 8; PERMANOVA, $p = 0.001$, $R^2 = 0.02$) (Fig. 3). In addition, in both systems, the OTU' co-occurrences (e.g., positive links) outranked the co-exclusions (e.g., negative links), with plantations showing 10,258 positive vs 2,270 negative links (ratio 4.5:1) and forests showing 7,918 positive vs 1,054 negative links (ratio 7.5:1). As regards of *T. melanosporum* centrality, its position in the network metrics' ranking was 276[th] in plantations and 58[th] in forests, i.e., within the first 38 % and 9 % network influential OTUs respectively, while this position shifted to 35.8 % and 21.2 % respectively when only the hub score was taken into account. Given that, although in neither truffle-producing system it could be characterised as a hub species, it can be suggested that *T. melanosporum* likely had a more central role in forests than in plantations.





**Figure 3: Soil fungal networks in the brûlé of black truffle producing plantations (left) and forests (right). For each truffle producing system, the total fungal network with co-occurrences (blue-positive links) and co-exclusions (orange-negative links) between OTUs (a) and the subnetwork composing of *Tuber melanosporum* with other fungal species (b) are depicted. Each node represents one fungal OTU, among which *Tuber melanosporum* is highlighted. The node size is related to the hub score index, and the node colour indicates a separate module inferred by Gephi (different colours were assigned to module IDs for each system).**



*Tuber melanosporum* was connected either positively (co-occurrence) or negatively (co-exclusion) with 2.3 % and 2.8 % of the total fungal OTUs in plantations and forests, respectively (Fig.3; Table S3). One third of these connections in plantations and more than half in forests represented co-occurrences of *T. melanosporum* with other fungi, while the rest of links represented co-exclusions (Table S3), with the ratio of the number of positive/negative links being approximately 0.4 and 1.25 in plantations and forests, respectively. Among these links, *Trichothecium crotocinigenum* (Schol-Schwarz) Summerb. was the only fungal OTU co-occurring with *T. melanosporum* in both producing systems (Table S3). Different *Mortierella* OTUs showed links with *T. melanosporum* in the co-occurring networks, both in plantations (negative link), and in forests (positive link). Particularly in plantations, *T. melanosporum* showed negative links with saprotrophic OTUs, both positive and negative links with OTUs characterised as plant pathogens, a negative link with a parasite and a positive link with one AMF OTU (Table S3). On the other hand, in forests, *T. melanosporum* was linked either positively or negatively with saprotrophic, pathogens, parasites and ECM OTUs (Table S3).

The role of black truffle structuring the soil fungal network was further examined by regression analysis, which revealed that fungal network complexity was reduced as *T. melanosporum* abundance increased, but only in plantations and not in forests (Fig. 4).

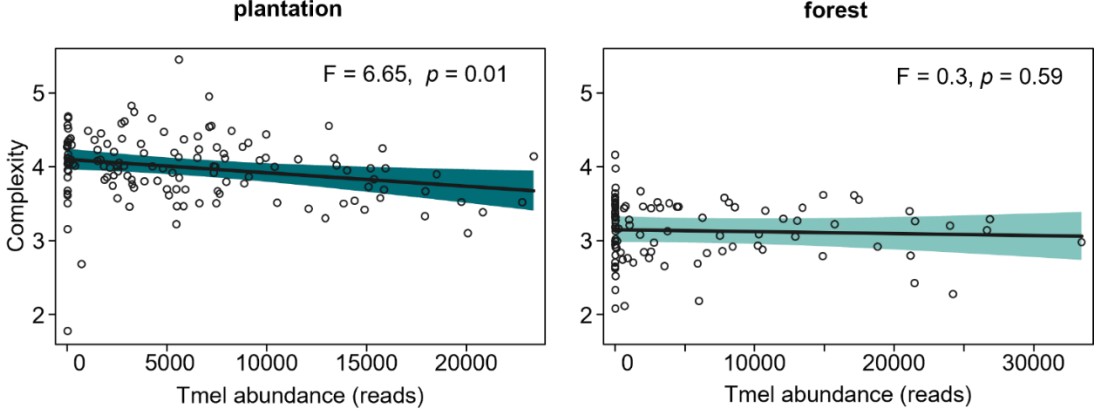

**Figure 4: Relations (mean predicted values) between *Tuber melanosporum* (Tmel) abundance and the soil fungal network complexity in black truffle producing plantations ($N = 135$) and forests ($N = 96$). The relationships were estimated by linear mixed models, with the site as a random factor. Tmel abundance measured as sequencing reads; network complexity was calculated as the ratio of fungal links by the OTU richness. The shaded area represents the 95 % confidence intervals.**

### 3.2 Ecological fungal guilds and soil functioning

Approximately 54 % of fungal taxa in the networks of both plantations ($N = 395$ out of 728 OTUs in total) and forests ($N = 346$ out of 641 OTUs in total) were assigned to an ecological fungal guild, representing 69 % and 73 % of the total fungal abundance in the respective datasets. In plantations, the relative abundance of saprotrophic (F = 25.9, $p < 0.001$), but also of plant pathogens (F = 45.1, $p < 0.001$), was significantly higher than in forests (Fig. 5). As expected (hypothesis 3), a greater abundance of ectomycorrhizal OTUs (other than *T. melanosporum*) was found in forests than in plantations (F = 88.7, $p < 0.001$).



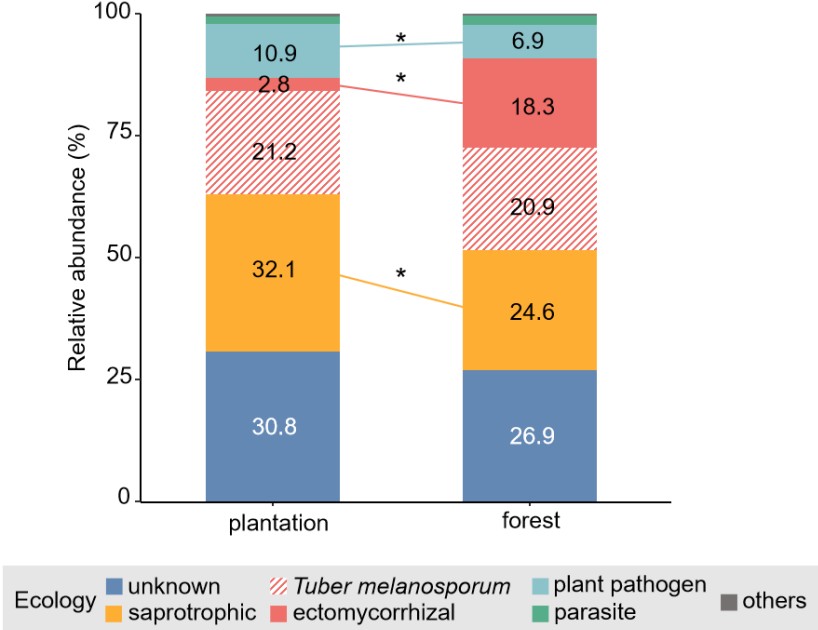

**Figure 5: Relative abundance of the main ecological fungal guilds, together with *Tuber melanosporum*, found in each truffle producing system. For a given fungal guild, asterisk represents significant differences between plantations and forests, derived from linear mixed models, with the site as a random factor. "Others" includes fungi classed as arbuscular mycorrhizal fungi, endophytes, epiphytes and lichens.**

To test if soil ecological fungal guilds could explain soil carbon and nutrient cycling, the ENET model was performed with the most representative guilds, separately for each type of truffle producing system (Fig. 6). In plantations, saprotrophs did explain both positively (C-related β-glucosidase, β-cellobiohydrolase, β-xylosidase, laccase, and N-related leucine-aminopeptidase) and negatively (P-related alkaline phosphatase, and N-related chitinase) the soil enzymatic activities (Fig. 6). By contrast, ECM fungi had a scarce effect explaining soil enzymatic activities. However, plant pathogens did have an overall negative contribution to many of the soil enzymatic activities measured. On the other hand, in forests, saprotrophs had a significant positive effect explaining most of the soil enzymatic activities, particularly cellobiohydrolase related to hemicellulose degradation (Fig. 6). ECM fungi did also explain positively most of the soil enzymatic activities (Fig. 6).





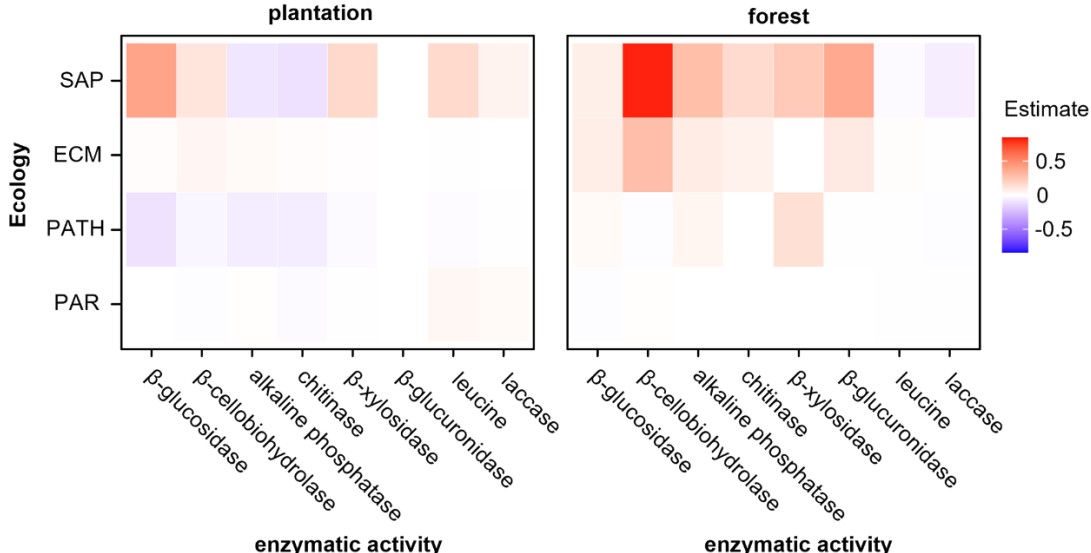

**Figure 6: Contributions of ecological fungal guilds to soil extracellular enzymatic activity in truffle-producing systems (plantations and forests). Heatmaps show the summed estimated values (positive in red and negative in blue) of fungal OTUs for each fungal guild, explaining at least one soil enzymatic activity by ENET models. SAP = saprotrophs, ECM = ectomycorrhizal fungi, PATH = plant pathogens, PAR = parasites. The estimated values are 0.8-scaled.**

## 4 Discussion

### 4.1 Diversity of soil fungal communities in wild and managed truffle producing ecosystems

In this study, we assessed the fungal community composition of black truffle brûlés in two types of truffle-producing systems (plantations and forests) and two sampling seasons (spring and autumn). Contrary to our first hypothesis, soil fungal richness was similar in plantations and forests, and between seasons. Although it is usually reported that soil fungal richness decreases when land use intensity increases (Bagella et al., 2014; Brinkmann et al., 2019), the opposite pattern or even unclear changes have also been observed in an extensive literature review (Balami et al., 2020). A study in *Quercus*-dominated habitats found a strong association of fungal richness with tree species richness (Saitta et al., 2018), and this could explain the similarity of fungal richness in our sampled sites, where *Q. ilex* was the dominant tree species both in plantations and forests. However, in our regional-scale study, soil fungal β-diversity did show significant differences between the type of truffle producing system and the seasons. A recent study of soil microbial diversity across Europe has identified land-use perturbation, climate, soil properties and vegetation cover as the main influencing factors on soil fungal diversity (Labouyrie et al., 2023). Our results agree with the differences in β-diversity of soil fungal community previously observed in Mediterranean and temperate climate sites, sampled across four seasons (Piñuela et al., 2024), although in the latter case no seasonal effects were detected, probably due to the limited number of plots included compared to our study.



Despite the spatial and seasonal differences in our studied sites, soil fungal communities were principally composed of
Ascomycota, in agreement with previous works (Fu et al., 2016; Herrero de Aza et al., 2022), while some of the most
abundant genera, apart from *Tuber,* were *Mortierella*, *Solicoccozyma*, *Fusarium* and *Tomentella*, similarly as detected in
other studies (De Miguel et al., 2016; Herrero de Aza et al. 2022). According to Egidi et al. (2019), the dominance of
Ascomycota in soil fungal communities is explained not only by their dispersal ability, but also their lifestyles and
colonisation ability across multiple niches.

### 4.2 Soil fungal networks in black truffle-dominated systems

In our study, the fungal co-occurrence network across all sampled black truffle brûlés showed a strong dependency on the
type of production system rather than the sampling season, which is consistent with the results of Piñuela et al. (2024).
Network links and complexity were higher in plantations than in forests, which may indicate a more stable fungal
community in the former system (Griffiths and Philippot, 2013). Recently, Byers et al. (2024) detected that the most densely
connected networks are found in land uses under intensive agricultural management, or under naturally regenerating
vegetation, rather than in native forests. However, contradictory results regarding soil microbial network complexity and
land use intensity and disturbance have been observed in other studies. For example, the conversion of a tropical rainforest to
rubber plantation increased the fungal network complexity (Lan et al., 2022), while the opposite occurred with the
conversion of a deciduous forest to a coniferous plantation (Nakayama et al., 2019). Zhao et al. (2024) also identified soil
fertility -followed by climate- as the primary driver influencing the complexity and stability of soil fungal networks in
Mongolian pine plantations. However, they cautioned that the interpretation of network outputs should be approached
carefully, as additional factors such as local environmental conditions or plant community composition may also influence.
Different ecological processes can potentially contribute to changes in co-occurrence patterns of communities at the regional
scale, including habitat filtering, competition or neutral processes (Horner-Devine et al., 2007; Maaß et al., 2014).
Differences in soil fungal network structure among forest and plantations have been, however, regularly explained by the
variation in soil properties (Wang et al., 2024b), or different vegetation cover, and subsequently in rhizosphere conditions
(Nakayama et al., 2019; Tang et al., 2024) (i.e., habitat filtering). Further studies are needed to disentangle the specific
ecological processes regulating the biotic (and abiotic) interactions in truffle-dominated ecosystems.

Once *T. melanosporum* is well established within the brûlé, it dominates the soil acting as a strong environmental filter for
plants and soil microbial communities (Napoli et al., 2010; Streiblová et al., 2012). Barou et al. (2025) showed that *T.
melanosporum* abundance negatively correlated with that of other fungi in wild and managed truffle-producing systems, and
with the overall fungal richness in plantations. But surprisingly in this study, despite its dominance in the brûlé, *T.
melanosporum* did not appear as hub species (i.e., did not potentially organize community-scale processes of microbial
and/or plant interactions in soil, Toju et al., 2018), neither in the whole soil fungal network nor in those separated by
plantations and forest. Instead, rather saprotrophs and plant pathogens had a central role in the soil fungal network. In fact, in
both truffle-producing systems studied, only a very small proportion of taxa of the total fungal community interacted with *T.




*melanosporum*. However, in plantations, soil fungal network complexity was negatively impacted by the abundance of black truffle, indicating that although it does not play a prominent role as a hub species, it probably exerts other indirect effects on the structure of the network. In addition, the prevalence of co-exclusions compared to co-occurrences of *T. melanosporum*

with other fungi in plantations indicates the competitive effect of the fungus in this productive system, which may consequently affect the whole fungal network. To confirm this, further research employing pairwise experimental designs that include samplings in/out of the black truffle dominated areas, i.e., the brûlés, would be needed.

Among the most abundant genera, only different *Mortierella* OTUs were related with *T. melanosporum* in the co-occurring networks, both in plantations (negative link), and in forests (positive link). Broad geographic and host-plant ranges

characterize this saprotrophic fungal genus, which often does not play determinant topological roles in local or regional plant-fungus networks (Toju et al., 2018). *Mortierella* species have been also reported to collaborate with AMF in phosphorus acquisition (Osorio and Habte, 2001; Tamayo-Velez and Osorio, 2017). In addition, the only common OTU linked with *T. melanosporum* in both type of production systems was *Trichothecium crotocinigenum*. According to the fungal trait databases, this fungus was assigned as plant pathogen (primary lifestyle) and litter saprotroph (secondary

lifestyle), but it has also been described as an endophytic fungus with antipathogenic effects in potato crops (Yang et al., 2018; Yang et al., 2020). Further research is needed to understand the positive association of *T. melanosporum* with this fungal species, as well as the interactions with *Mortierella* species within the brûlé.

### 4.3 Functional contribution of soil fungal guilds in the brûlé

Concerning forests and managed truffle-producing systems, significant differences in the abundance of most soil fungal

guilds were observed. Saprotrophs and plant pathogens were significantly more abundant in plantations than forests, while the opposite happened for ECM fungi other than *T. melanosporum*. Since in plantations the host trees had been initially inoculated with *T. melanosporum* in nurseries, this fungus probably had priority advantage over other ECM species to colonise new root-tips of the host trees (Kennedy and Bruns, 2005). Additionally, soil legacies can play a role since plantations are established in previous agricultural lands usually dominated by saprotrophs, and devoid of ECM propagules

compared to forests (Phillips et al., 2013). It has been also reported that intensive land use shapes soil microbial communities and increases the abundance of potential plant pathogens (Idbella and Bonanomi, 2023).

When the different fungal guilds were further tested for their contribution in soil functioning, saprotrophs did significantly predict most of the soil enzymatic activities tested in both truffle-producing systems. A recent study in Mediterranean oak stands showed that saprotrophs were the main fungi responsible for high C and N mineralisation (Adamo et al., 2022); this

can be probably related to fast decomposition of high-quality litter by saprotrophs, releasing nutrients in the mineral soil that can be mined by other microorganisms, e.g., mycorrhizal fungi (Fernandez and Kennedy, 2016; Lebreton et al., 2021). It cannot be ruled out that other soil saprotrophs as bacteria can be facilitating N cycling in soil (Phillips et al., 2013). In addition, plant pathogens were negatively related to most soil enzymatic activities in plantations, where their abundance was higher than in forests. Contrary to woodlands, grasslands and croplands are related to a rapid turnover of nutrients and



organic matter (Phillips et al., 2013), an easier nutrient leaching out, and a higher soil-borne potential fungal plant pathogens abundance (Romero et al., 2024). Thus, these ecosystems have been proposed to be more reliant than forests on soil health (i.e., capacity of soils to continuously support plant productivity and other ecosystem functions such as nutrient cycling; Lehmann et al., 2020) to regulate their productivity. Finally, ECM fungi were significantly associated with soil enzymatic activity, especially in forests. Recently, Prieto-Rubio et al. (2024) showed that the structure of ECM fungal networks

predicted soil enzymatic activities in Mediterranean pine-*Quercus* mixed forests, although taxa acting as keystone within the fungal networks were not major determinants of soil functioning. In this sense, the association of ECM with the enzymatic activity in forests in our study could represent the indirect effect of *T. melanosporum* on the network structure. In any case, the extent to which the dominant effect of *T. melanosporum* is synergistically linked to the dynamics of saprotrophs and/or plant pathogens in plantations, as well as ECM in forests, and the potential functional consequences of these interactions,

still requires much more investigation.

## 5 Conclusions

In this study, we aimed to define the structure of soil fungal communities associated with black truffle producing plantations and forests, and to decipher the relationship between fungal guilds and carbon and nutrient cycling in soils dominated by *T. melanosporum*. To our knowledge, this is the first study exploring the co-occurrence fungal network structure in black truffle

brûlés and identifying fungal links with *T. melanosporum*. The soil fungal communities were more homogeneous (i.e., more similar taxa) and showed a more complex network structure in plantations than in forests, indicating more stable fungal communities, probably related to the uniform environmental conditions and reduced plant community diversity in monoculture plantations. *T. melanosporum* had a more central role in the fungal network of forests than in plantations, but in neither system was a hub species despite its dominance in the brûlés, while it showed only a few links with other fungi,

mainly with saprotrophic and plant pathogens taxa. Among the ecological fungal guilds, mainly saprotrophs and partially ECM and plant pathogens explained the carbon and nutrient cycling in both truffle-producing systems. Understanding the ecology, including the biodiversity and functioning, of black truffle-dominated soils will provide a stronger foundation for making informed decisions regarding the management of black truffle plantations.

## Data availability

Data will be made available from the corresponding author upon reasonable request.



## Author contributions

Vasiliki Barou: Writing – review & editing, Writing – original draft, Visualization, Validation, Software, Methodology, Formal analysis, Data curation. Jorge Prieto-Rubio: Writing – review & editing, Validation, Software, Methodology. Mario Zabal-Aguirre: Writing – review & editing, Validation, Software, Methodology, Data curation. Javier Parladé: Writing –
review & editing, Validation, Supervision, Resources, Project administration, Methodology, Investigation, Funding acquisition, Conceptualization. Ana Rincón: Writing – review & editing, Validation, Supervision, Resources, Project administration, Methodology, Investigation, Funding acquisition, Conceptualization.

## Competing interests

The authors declare that they have no conflict of interest.

## Acknowledgements

We gratefully acknowledge all the collaborating stakeholders for their help in defining black truffle productive trees and locations. We also thank B. Pallol, A. Vázquez and A. Romero for their help in the lab work.

## Financial support

The work was supported by the projects TUBERSYSTEMS (RTI2018-093907-B-C21/22) and TUBERLINKS (PID2022-
1364780B-C31) founded by the Spanish Ministry of Science and University (MCIU). This research is part of the thesis of the first author, V. Barou, who was enrolled in the program of Plant Biology and Biotechnology of Universitat Autònoma de Barcelona and held a pre-doctoral fellowship awarded by MCIU (PRE2019-091338).

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
