# Peer review of "Soil fungal network complexity and functional roles differ between black truffle plantations and forests"

_EGUsphere, 2025_

## Author Comment (AC1)

Tittle: **Managed black truffle-producing systems have greater soil fungal network complexity and distinct functional roles compared to wild systems**

We sincerely thank the Editor for the opportunity to revise our manuscript and the Reviewer for their thoughtful and constructive comments. We are encouraged by the positive assessment of our work and appreciate the suggestions, which have helped us improve the clarity, precision, and overall quality of the manuscript.

As indicated by the editor, we have now modified the end of introduction to make it clearer the objectives and hypotheses of our work.

In response, we have carefully addressed each point raised and made the necessary revisions throughout the text, as indicated in our response. We note that while several improvements have been made, the main findings and conclusions of the study remain unchanged.

We trust that the revised version now meets the requirements for publication, and we hope the manuscript will be found suitable for acceptance.

———————

**Reviewer_1**

**Summary:**

In this article (egusphere-2025-2078), the authors set out to investigate the role of black truffle (*Tuber melanosporum* Vittad.) in shaping the fungal community in the soil ecosystem it grows in. They focused solely on the soil fungal community (not considering prokaryotes) and compared the more "natural" forest soil system against the cultivated plantation system. They also compared samples from spring and autumn to gain insights into seasonal effects on the role of *T. melanosporum*. Based on previous studies and general knowledge of fungal ecology, the following hypothesis were put forth:

      1. a) soil fungal networks are richer and more complex in forests compared to plantations.
        b) a differential seasonal effect on soil fungal communities can be observed
      2. *T. melanosporum* is strongly connected in the soil fungal network, possibly acting as a hub species
      3. a) the prevalent functional fungal guilds differ when comparing forest and plantation systems
        b) greater prevalence of ectomycorrhizal fungi vs saprotrophs can be observed in forests compared to plantations

To investigate these hypotheses, topsoil (0-20 cm) from inside the brûlé, i.e. presumably area affected by *T. melanosporum*, was obtained from both systems (four replicates) and in both seasons. The brûlé boundaries were determined visually and multiple samples from within the area were combined to a composite sample, but no negative controls from outside

that area were taken. Some sampling sites were paired (forest and plantation in close proximity), while others were not. Fungal occurrence in the samples was determined by metabarcoding. Co-occurrence networks were created based on this and the role of *T. melanosporum* within these networks was studied. Soil functioning was proxied through the potential activities of eight exoenzymes related to carbon (β-glucosidase, β-cellobiohydrolase, β-xylosidase, β-glucuronidase, and laccase), nitrogen (chitinase and leucine-aminopeptidase), and phosphorus (alkaline phosphatase) cycling. These were measured for the soil samples and the potential role of different fungal guilds in explaining these activities was predicted by modeling.

Forest fungal communities showed significantly greater β-diversity, while α-diversity did not differ significantly between plantation and forest. Based on a single mixed co-occurrence network, OTU links and network complexity appeared significantly higher in plantations compared to the forest system (contrary to expectation from hyp. 1), while no significant difference between seasons was observed. In separately modeled co-occurrence networks for both ecosystems, *T. melanosporum* was not strongly connected to other OTUs and did not appear to act as a hub species (contrary to hyp. 2). Differing abundance of fungal guilds was observed between both systems and ectomycorrhizal were more prevalent in the forest (fitting hyp. 3).

We thank to Referee #1 for his/her insightful and encouraging assessment. We have answered point by point to his/her comments in blue, as follows:

**Key limitations of the study:**

1. No control samples outside the brûlés were taken, meaning there was no true negative control. The authors themselves identify this limitation (l. 382 – 387), but do not sufficiently address it in their analyses. Co-occurrence works by checking shared patterns of presence or absence. Since only samples from truffle-dominated areas were used which would be expected to almost always contain *T. melanosporum* reads, positive connections would only be expected with other highly abundant taxa, since only these could match the truffles occurrence pattern. Negative connections would also not hold as much informative value in this specific sampling approach, since they would likely mainly depict less abundant / more rare taxa that occur in few samples.

*Tuber melanosporum* was the most abundant species in all samples (see also Fig. S2 of Barou *et al.*, 2025, where the same study area and samples were assessed). However, an abundance gradient was observed across samples allowing us to explore variations in the relationship with other fungi. We agree that there was no true negative control and future studies including samples in- and outside the brûle are needed to complement our results, which has been now pertinently highlighted in the discussion and conclusions.

Regarding the network analysis, the applied algorithm SPIEC-EASI relies on transformed data, and its output is the inverse of the covariance matrix, not a correlation matrix. The relative abundances of the OTUs are compositional, meaning they are constrained to sum to a constant (usually 1 or 100%) which introduces a mathematical dependency between components—if an OTU´s abundance increases, others´ abundance must decrease.  It does not imply that if one fungus increases, another increases or decreases. To better clarify this, we added some more details in section 2.3 of Material and Methods. The resulting matrix

from SPIEC-EASI allows us to detect positive values, i.e., positive conditional dependence (co-occurrences); negative values, i.e., negative conditional dependence (co-exclusions); or null values, i.e., conditional independence (no edges in the network graph). In this sense, the co-occurrence network-based approach has given us the opportunity to analyse coexistence patterns among fungal species within the brûlé (in natural or managed systems), meaning that the experimental design assumes the presence of *T. melanosporum* in all samples.

One reason for *T. melanosporum* not showing up as a hub species in the analysis could also be that it already modified the microbiome and reduced the abundance of some other fungi.

We agree with this comment, and this limitation has been now emphasized in 4.2 section of the discussion.

Without the outside control, we are unable to compare to the "undisturbed" ecosystem without truffle dominance, which really limits what can be deduced about its actual role in the system.

There is a clear dominance of *Tuber melanosporum* and an abundance gradient of the fungus across samples, which to some extent allows us to assess its effect on the soil. However, the lack of control outside the brûlé and the necessity for new studies has been pertinently emphasized in the 4.2 section of the discussion and also in conclusions. Our objective was to investigate the close environment of *T. melanosporum*, distinct from examining microbiome shifts within and outside the brûlé caused by *T. melanosporum* itself. Specifically, we sought to determine the positive and negative fungal associations with *T. melanosporum*, which required focused sampling within the brûlé because identifying the species associated with the black truffle can be crucial in designing truffle plantation management practices.

In a similar vein, the negative control would have also allowed researchers to rule out environmental filtering as the main driver for co-occurrence, by depicting the community without truffle but in the exact same soil conditions. Some of the sampling sites appear at least somewhat paired, while others are completely singular, which makes it hard to disentangle the actual effect of *T. melanosporum* on the local soil microbiome, compared to differences purely based on abiotic factors. This issue could maybe be circumvented by comparing matched sampling sites. While there can be merit in combining all the data to finder larger underlying trends, some nuance will inevitably get lost by lumping these potentially diverse and unbalanced datasets together.

We agree, and it is precisely to avoid possible environmental confounding factors why we decided to separate network analyses for forests and plantations. Furthermore, the discussion highlights the significant potential of pursuing this analytical approach in future studies (section 4.2), a direction that is already being explored in a complementary ongoing study.

Since detailed per-sample soil parameters are not supplied to reviewers, it is difficult to decide whether this would have been a sensible measure.

In our previous work, Barou et al. (2024), we provided both the methods used to perform soil physical-chemical analyses and the mean ± SE values of a range of soil parameters for plantations and forests, as it is referred in the 2.1 section of Materials and Methods, and also summarized in the present study (see supplementary Table S1).

2. *T. melanosporum* is described as being "dominant" in the brûlé (l. 31 – 33), leading to the distinct and visible vegetation pattern, which also formed the basis for picking sampling spots. Based on this one would expect it to be found in almost every sample, especially since 4 subsamples from each tree were combined. However, fig. 4 shows that for plantations some and for forests a lot of samples appear to have ~0 Tmel reads.      This data is only presented as a plot, no table with the exact numbers (sample number + number of Tmel reads) is provided, but based on the figure it seems like a decent chunk of the soil samples per brûlé did not contain any *T. melanosporum* DNA, or not enough to be detected by the metabarcoding approach. This raises the question whether a simple phenotypic determination of truffle-dominated soil is sufficient for actually picking positive samples, or whether amplification efficiency of the ITS region is sufficient. In the current version of the manuscript, the authors do not address the zero Tmel abundance samples at all, which would be a critical point to discuss.

We acknowledge that in the figure, samples with low abundance may appear to have zero reads. However, this is a result of the large scale used in the figure (ranging from 0 to 20,000), which visually compresses the lower values. In fact, no sample had zero Tmel reads; rather, some samples had low read counts, i.e. ≥ 4, as it has been now indicated in the legends of Figs. 1 & 4. In fact, the mean number of Tmel reads per truffle system exceeded 5,600, confirming consistent detection across all samples in both truffle-producing systems. To avoid any confusion, attention to the lack of zero Tmel has been now drawn in the results section (3.1 section). In addition, a piece of discussion has been added to acknowledge the potential limitations pointed by the reviewer (in 4.2. section of the discussion), with which we totally agree.

3. Only the fungal perspective is considered, despite bacteria likely making up a large part of the soil microbial community, especially at the alkaline pH found at the sampling sites. This means that only the interactions with the small fungal subset of the soil microbiome are considered. While additional amplicon sequencing for bacteria would have likely exceeded the scope of this study, some less complex methods like a comparison between general 16S vs ITS qPCR could at least have helped quantifying how much of the overall community is not included in this analysis.

We fully agree, and in fact, this is the natural progression of our ongoing work. It would also be incredibly interesting to investigate the functional implications of the complete trophic network (i.e., including fauna) within these unique systems. This will be the focus of future research by our group.

**Conclusion:**

We understand that some of the mentioned limitations are hard to address without extensive resampling or sequencing, but we strongly urge the authors to reconsider which conclusions can be drawn from their data and which questions go beyond their scope. Especially the title

(l. 1 – 3) as well as the statements about a stronger negative influence of black truffle on the fungal network in plantations (l. 382 – 387) should be carefully reevaluated and potentially rephrased.

Following the recommendations, we have now tone down the paragraphs where potential negative influence of *T. melanosporum* was addressed all along the manuscript (e.g. "spreads", "colonising", "presence" instead of "dominates", "dominating" and "dominance" in the Introduction and Discussion sections). For consistence with this, the title has been also re-evaluated: "Soil fungal network complexity and functional roles differ between black truffle plantations and wild-producing forests".

Without the negative controls that would depict the undisturbed network, these conclusions do not just require confirmation but lack strong proof altogether, especially since members of the community that might have been fully suppressed by *T. melanosporum* are not accounted for here. An approach of only using paired sites to counteract some of the study design limitations could be promising to investigate the influence of abiotic conditions, as well as forest vs plantation on the fungal community. The issue of brûlé samples without any detectable *T. melanosporum* reads should also be further investigated and put into the focus of the revision.

We sincerely thank the Reviewer for this comment. We agree that clarifying these specific research questions strengthens the novelty and scientific contribution of our work. Following the advice, we have revised the entire discussion section to more explicitly emphasize the key points raised. Soil brûlés have been considered as the best indicator of *T. melanosporum* presence because of the well-known allelopathic effects of truffles on the surrounding vegetation (Streivlová et al. 2012). Also, the size of the brûlé has been related with truffle productivity (García-Montero et al 2007). However, in some cases other soil fungal allelopathy may produce similar effects on plant growth as those observed in the truffle brûlés (Osivand et al. 2018). This could explain why some well-formed brulés show a very low amount (but always above 0) of *T. melanosporum* reads in our study.

In any case, the varying levels of Tmel abundance in well-developed brûlés in our study raise relevant questions about whether a simple phenotypic assessment of truffle-dominated soil is sufficient to identify positive samples, whether higher pooling efforts in soil sampling and/or DNA extraction should be done to minimise soil spatial/temporal heterogeneity or if amplification efficiency of the ITS region alone is adequate, issues that warrant further investigations. As indicated, all these limitations and the continuity of the work have now been thoroughly addressed in the revised version of the manuscript. The need of additional studies and designs has been also highlighted in the discussion and the conclusion sections.

**References:**

Streiblová E, Gryndlerová H, Gryndler M. Truffle brûlé: an efficient fungal life strategy. 2012-FEMS Microbiol Ecol. 80(1):1-8. doi: 10.1111/j.1574-6941.2011.01283.x

García-Montero, L.G.; Manjón, J.L.; Pascual, C.; García-Abril, A. 2007. Ecological patterns of *Tuber melanosporum* and different *Quercus* Mediterranean forests: Quantitative production

of truffles, burn sizes and soil studies. For Ecol Man 242 (2-3): 288-296. doi: 10.1016/j.foreco.2007.01.045

Osivand, A.; Araya, H.; Appiah, K.S.; Mardani, H.; Ishizaki, T.; Fujii, Y. Allelopathy of Wild Mushrooms—An Important Factor for Assessing Forest Ecosystems in Japan. 2018. Forests *9*, 773. https://doi.org/10.3390/f9120773

---

## Author Comment (AC2)

**Reviewer_2**

The article "Managed black truffle-producing systems have greater soil fungal network complexity and distinct functional roles compared to wild systems" by Barou et al. characterizes the fungal community in truffle plantation soils of Spain (with an outstanding sample size) and compares it with the fungal community in the soil of wild truffle-producing forests, while providing in-depth insight into the role of Tuber melanosporum in the soil fungal community networks, as well as into the influence of the most important fungal guilds on soil enzymatic activity. Truffle cultivation has traditionally advanced by observing the ecology of truffle in the wild. However, comparisons of wild and cultivated truffle sites are rare. This makes the study relatively novel. Besides, the information on soil enzymatic activity of the truffle soils is also relatively novel, even more its study in relation to the composition of the soil fungal community. All this information is an incremental knowledge that helps to better understand the role of truffle within the soil fungal community of truffle plantations and to unravel the role of this fungal community into the productivity and sustainability of truffle cultivation. The introduction of the study provides a comprehensive review of the state of the art, as well as clear and specific objectives and hypothesis for the study. The materials and methods are sufficiently described. The results are clearly exposed and provide an exhaustive analysis of the data, delving into ecological relationships with potential agronomic interest. The discussion connects the study objectives and hypotheses with the results, thoroughly exploring the ecological and practical implications of the study.

We thank to Referee #2 for his/her insightful and encouraging assessment. We have answered point by point to his/her comments in blue, as follows:

However, a paragraph with the study limitations, the potential lines for future research and/or the practical implications of the study for truffle cultivation could help emphasizing the relevance and novelty of the study ("Understanding the ecology, including the biodiversity and functioning, of black truffle-dominated soils will provide a stronger foundation for making informed decisions regarding the management of black truffle plantations" may not be the strongest sentence to end the manuscript).

We greatly appreciate this suggestion. Following the recommendations, we have better elaborated the closing sentence of the manuscript by including the practical implications of the study. The new ending for the conclusions paragraph is "*Further research based on pairwise experimental designs that include samplings in/out of the black truffle dominated areas, i.e., the brûlés, would help us to clarify the biotic and abiotic transformations induced by T. melanosporum in soils. The findings of this study on fungal biodiversity and the functioning of black truffle-colonised soils offer valuable practical insights, providing a robust scientific foundation to enhance decision-making and drive more effective management strategies, such as those related to fertilisation and soil microbiome management, for black truffle plantations.*"

However, a few minor issues should be clarified:

1) L125 Which were the dates of sampling? Besides, the final sampling size is 231 (explained in L137), but it is not clear which is the final size for wild/cultivated and for spring/autumn.

The samplings were conducted during October and November 2019 for the autumn campaign, and during April and May for the spring campaign.
For both autumn and spring campaigns 68 trees were sampled in plantations and 48 trees in forests as mentioned in L125, which made a total of 232 samples processed. However, due to a failure in a plantation sample of spring, 231 samples were finally considered for the analysis. We have now added this detail about the plantation-spring sample that failed in the manuscript at the 2.2 section of Materials and Methods.

2) L129 "Soil functioning was proxy through the potential activities" Is this sentence ok?

We have rephrased the sentence for a better understanding, as follows: "As a proxy for soil functioning, the potential activities of eight exoenzymes related with… were calculated." (2.1 section of Materials and Methods).

3) L189 The tests for hypothesis 2 do not seem to correspond with the hypothesis 2 specified in L101, since no hypothesis talks about soil parameters. Besides, the study aims to find the differences in the fungal community between cultivated and wild truffières (L91). For this, the networks of both types should be compared, or alternatively, it should be tested whether the network for wild sites is different from a random network and whether the network for plantations is different from a random network.

Thank you for this observation. In fact, our second hypothesis is whether *T. melanosporum* is a hub species within the fungal network of plantations and/or forests.
In L189, the impact of the type of truffle productive system and season on the fungal network is linked to hypothesis 1, not to hypothesis 2 as previously stated leading to the misunderstanding pointed by the referee, and we have now corrected this in the revised version of the manuscript.
We also have now restructured the hypotheses at the end of the Introduction to ensure clarity and to explicitly include all the factors and variables analysed in the study.

4) L205. How does ENET methodology deal with proportion data (percentage of reads for a guild)? Proportion data are frequently not normal data (GLMM?). Besides, since 3-4 guilds (including "non classified OTUs") practically dominate the community, the percentage of the main guilds are most likely highly correlated. How do you assess that colinearity does not affect the results of ENET methodology?

Before running ENET, we scaled the OTU matrix to the minimum sequencing depth, filtered out the least abundant OTUs, and then standardized with *decostand* function, allowing for equitable comparison of OTUs. As we mentioned in L206-209, the LASSO and Ridge penalty-based regression modelling incorporated in ENET (Zou and Hastie, 2005) helps to avoid the overfitting and the collinearity between OTUs.

5) L239 Figure S4 seems closely related to the main specific objectives. Why is it not included in the main manuscript? Besides, why did you decide to characterize the alpha-diversity only with richness and not with indices of diversity such as Shannon or Simpson?

The Figure S4 is indeed related with our first hypothesis. We considered presenting Figure 1 in the main text because we thought that it would be useful to have a resume of the fungal composition of our samples at the begging of the results section, and then to introduce the results relative to the first hypothesis. Since this hypothesis has many components, we decided to keep Figure 2 in the main text, as it presents novel results, and move Figure S4 to the Supplementary Material in order to avoid overcrowding the manuscript.
On the other hand, there are many alpha-diversity indexes but, for simplicity, we decided to use just richness in our analyses.

6) L240-241 Significant differences in the PERMANOVA can be related to both differences in the centroid location and in the dispersion of each group samples, which seems to be the case according to Fig. S4b. Contrary to what is said in the manuscript, and according to Fig. S4b, the communities do not seem clearly dissimilar, but only different in dispersion.

Thank you for your observation regarding the apparent discrepancy between the PERMANOVA results and the NMDS plot. We agree that the visual separation in the NMDS is not striking, and this can be primarily due to the inclusion of the factor *site* by the strata fucntion in the PERMANOVA analysis. This is now mentioned in section 2.4 of the Materials and Methods, for clarity.

Through the strata function, PERMANOVA accounts for site variation —in this case, controlling for site-level differences similar to a random factor in LMM—when testing for the significance of our fixed factors type and season. This allows the model to detect subtle but consistent shifts in community composition across groups, even when those shifts are not visually prominent in unconstrained ordination methods like NMDS.

NMDS, by contrast, does not incorporate the strata function or control for site-level variation. It represents overall dissimilarity patterns, which may be dominated by site effects or other sources of variation not related to the factors of interest. As a result, the visual clustering in NMDS may not totally reflect the statistical significance detected by PERMANOVA.

7) L243-244. Wouldn't it be more correct to say that plantations tended to show higher values of pH, K and active carbonate?

While indeed the values of pH, K and active carbonate in plantation samples are higher than in those from forests (it can be also observed in Table S1), the interpretation of the environmental vectors has to be focused on their association with the fungal community structure in the respective truffle producing systems. To make clearer the purpose of the *envfit* function that fitted the environmental vectors, we have now added a small explanation in the 2.4 section.

8) L244-245. Taking into account that both seasons are almost centered in the biplot center, wouldn't it be more correct to say that spring showed more extreme values of pH, OM, Fe, although not always in the same direction?

Yes, indeed. We have now corrected this sentence.

9) L305 Why did you use T. melanosporum No of reads for regressions and not relative abundance of T. melanosporum, which you previously chose as normalized variable (L149)?

We totally agree that *T. melanosporum* relative abundance is a normalized variable and probably is more ecologically meaningful than absolute counts. We applied linear models with this variable and, while the results were quite similar to those of the models done with the number of reads as explanatory variable (and they had almost the same plots), showing a significant relationship with network complexity only in plantation samples, the fitting of the models was not as good. Thus, we preferred to keep the models with the number of reads.

10) Careful with British/American English (e.g. normalised/normalized).

We have now corrected this inconsistency along the manuscript.

11) L321 "To test if soil ecological fungal guilds could explain soil carbon and nutrient cycling". L407 "When the different fungal guilds were further tested for their contribution in soil functioning, saprotrophs did significantly predict most of the soil enzymatic activities tested in both truffle-producing systems". Does the ENET provide correlation values or a partition of the variance? The manuscript suggests the latter. If so, which proportion of the variance in enzymatic activities do the fungal guilds explain?

The ENET is a "feature selection" method that considers all taxa simultaneously and selects those taxa (predictors) that achieve the best prediction of an ecosystem function. The model results in coefficients for each OTU, where non-zero coefficients are used to infer a positive or negative contribution of an OTU to improving ecosystem functioning (Wagg et al., 2019), which in our case is the soil enzymatic activity. Thus, the model does not provide an explained proportion of variance nor a correlation, but causal relations between fungal guilds and the soil enzymatic activity.

12) L346-347 "Our results agree with the differences in β-diversity of soil fungal community previously observed in Mediterranean and temperate climate sites, sampled across four seasons (Piñuela et al., 2024)". How do they agree?

In the study of Piñuela et al. (2024), the fungal community composition appeared more scattered in the samples of wild sites compared to the plantation ones in the NMDS plot (Fig. 1), similarly to our plot, where the forest samples showed a more dispersed fungal community than the plantation samples. We have now added a brief explanation of this result's agreement in the manuscript (4.1 section of the Discussion).

13) L356 Brûlé in italics?

We initially introduced the term "brûlé" in quotation marks to highlight it as a concept. Given that brûlé is a well-established and widely recognized term among truffle and soil fungal researchers, we chose not to italicize it throughout the manuscript.

14) L370 "Differences in soil fungal network structure among forest and plantations have been, however, regularly explained by the variation in soil properties (Wang et al., 2024b), or different vegetation cover..." Considering this, it would be interesting to discuss a little the differences in vegetation structure between wild/cultivated sites (age of trees, percent canopy cover, percent soil cover by litter, periodic tillage, percent soil cover by herbs/shrubs, etc.).

We agree and, as recommended, we have now added a piece of discussion considering this point at 4.2 section of the Discussion.

REFERENCES

Wagg, C., Schlaeppi, K., Banerjee, S., Kuramae, E. E., and Van Der Heijden, M. G. A. Fungal-bacterial diversity and microbiome complexity predict ecosystem functioning, Nat. Commun., 10(1), 4841, https://doi.org/10.1038/s41467-01912798-y, 2009.

Zou, H., and Hastie, T.: Regularization and variable selection via the elastic net, J. R. Stat. Soc. Ser. B Methodol., 67(2), 301–320, https://doi.org/10.1111/j.1467-9868.2005.00503.x, 2005.

---

## Author Response (AR2)

**RESPONSE TO REVIEWERS**

**Referee #1**

We are pleased to notice that this re-submitted version sufficiently addresses our major concerns. Mainly, the slightly overstated title and conclusions were toned down a bit, making them more aligned with the strengths and limitations of this research. The presented results, embedded in the greater context of the other studies of the same group, offer a good starting point for unraveling the role of *T. melanosporum* in the fungal community of black truffle brûlés. From this foundation, the experimental setup and scope of the analysis can be tweaked to further enhance the clarity of the results.

At this stage we do not find any additional major issues with this manuscript, but the text could really benefit from a more detailed explanation of the network analysis with the SPIEC-EASI algorithm. Luckily the authors already provided this in the direct reply to our comments. To enhance readability and allow others to follow this pivotal analytical step of the study, we urge to authors to include a concise version of that specific part of their reply in the manuscript. Some additional minor corrections and suggestions for phrasing are listed below.

l. 26: „dominance" - change for less strong word, maybe abundance?

l. 36: "the black truffle mycelium spreads the area around the colonised trees" – spreads is good correction, just needs a preposition to complete the sentence, eg. "spreads throughout"

l. 79: "species" - is your metabarcoding data enough to fully resolve to the species level? Maybe also refer to OTUs here.

l. 86: "nutrient's mobilization" – should simply be "nutrient mobilization"

l. 114: "lifestyle" – should be lifestyles

l. 155 "physical-chemical" – should be "physico-chemical"

l. 184 "fungal database nature"- hard to understand what this term means. "Nature of the fungal database?"

l. 402 "may also influence. " … the results / … the outcome

l. 412 see l. 36 – "spreads throughout"

l. 418 "…abundance negatively correlated with…" – should be "…abundance was negatively correlated with…"

l. 423 "…plantations and forest, probably because…" – either both "plantation and forest" singular or both plural

We thank the reviewer for the constructive feedback and helpful suggestions. We have carefully addressed all the points raised. Also, a concise explanation of the SPIEC-EASI algorithm and its interpretation has been incorporated into the 2.3 section of *Materials and Methods* to enhance clarity and readability.

**Referee #2**

In my opinion, the revision greatly improved the manuscript, which now reflects more faithfully what was done and what happened in the experiment. I would just point out two minor grammatical issues:

L154 analised

L412 it spreads the soil

We thank the reviewer for the helpful feedback. The two suggested grammatical corrections have been carefully implemented in the revised manuscript.

**LIST OF CHANGES**

| Point in revised manuscript | Description of change |
|---|---|
| L26 | Changed "dominance" to "abundance" |
| L36 | Added "throughout" after "spreads" |
| L79 | Changed "species" to "OTU" |
| L86 | Changed "nutrient's mobilization" to "nutrient mobilization" |
| L114 | Changed "lifestyle" to "lifestyles" |
| L154 | Changed "analised" to "analysed" |
| L155 | Changed "physical-chemical" to "physico-chemical" |
| L184 | Changed "fungal database nature" to "nature of the fungal database" |
| LL187-190 | The phrase "In the resulting networks…in all samples" was added to provide a concise explanation of the SPIEC-EASI algorithm and its interpretation. |
| L407 | Added "results" after "influence" |
| L416 | Added "throughout" after "spreads" |
| L423 | Added "was" after "abundance" |
| L426 | Changed "forest" to "forests" |